# Chitosan/Lactic Acid Systems: Liquid Crystalline Behavior, Rheological Properties, and Riboflavin Release In Vitro

**DOI:** 10.3390/ijms232113207

**Published:** 2022-10-30

**Authors:** Natalia M. Selivanova, Aliya I. Galeeva, Yuriy G. Galyametdinov

**Affiliations:** 1Department of Physical and Colloid Chemistry, Kazan National Research Technological University, Kazan 420015, Russia; 2Zavoisky Physical-Technical Institute, FRC Kazan Scientific Center of RAS, Kazan 420029, Russia

**Keywords:** lyotropic liquid crystals, chitosan, lactic acid, Casson flow model, riboflavin release

## Abstract

Chitosan or its derivatives exhibit lyotropic liquid crystalline mesophases under certain conditions due to its semi-rigid structures. This work describes the development of chitosan-based biocompatible systems that include new components: lactic acid and non-ionic surfactants. Polarized optical microscopy studies revealed that these systems are capable of forming gels or lyotropic liquid crystals (LLCs) in a certain range of chitosan and lactic acid concentrations. According to the viscosity studies, the rheological flow of the LLCs can be accurately described by the Casson flow model. The intermolecular interactions of the LLC components were studied by FTIR spectroscopy. According to the FTIR data, hydrogen bonding is supposed to be responsible for the formation of the LLCs. In the studied systems, this LLC complex exists as the [ChitH^+^·CH_3_-CH(OH)-COO^−^] ion pair. The studied gel and LLCs were shown to possess the most prolonged release capabilities for riboflavin among similar binary LLC systems. The supramolecular organization and rheological characteristics of the studied chitosan-based systems were found to affect the release of riboflavin.

## 1. Introduction

Drug delivery systems based on soft-matter components are currently attracting growing attention. In such systems, soft-matter carriers are as important as their drug loads. They can considerably improve the bioavailability of their therapeutic loads and protect healthy cells from the toxic effects of certain therapeutic agents [1]. Soft delivery systems offer additional advantages, such as improved solubilization, a controlled drug release rate, the protection of drugs from hydrolysis or other types of chemical degradation, and improved drug availability. In this respect, LLCs demonstrate a considerable potential as soft delivery systems [2]. LLCs that consist of amphiphilic molecules can self-assemble in solutions and form various organized media with unique microstructures and physicochemical properties. As drug delivery systems, LLCs can provide various options for drug administration including, oral, topical, transdermal, ocular, or intranasal routes [3]. Rajabalaya et al. [4] discuss the structural evaluation of LLCs and their effects in oral and transdermal drug formulations. The LLC structure influences drug localization, particle size, and viscosity, which, in their turn, determine drug delivery properties. The components of LLCs are biodegradable and biocompatible, especially those based on lipids [5] or biopolymers. Their nanoscale and occlusive nature allows the overcoming of the rate-limiting barrier and offers the best option for curing skin disorders. LLCs provide an improved permeation and skin retention in the stratum corneum and viable epidermis [6]. It facilitates the targeted delivery of therapeutics to the superficial layers of skin disorders. The fact of amenability regarding the surface functionalization of the LLCs favors their suitability for the active targeting and development of theranostic agents [3]. This research area has been actively developing over the last decade, and is widely discussed in the literature [7,8]. The structural similarity of lyomesophases to membranes makes it possible to consider LLCs as models of biological objects that are suitable for simulating the molecular structures and functions of various cellular organelles, tissues, and organs. Due to their unique properties and structural features, LLCs are capable of encapsulating a wide range of substances and transferring both hydrophobic and hydrophilic bioactive components, such as tetrapeptide [9], vitamins [10], or substituted calixarene [11]. These properties make them particularly attractive for drug delivery and carrying applications. In this regard, the synthesis of novel biocompatible components that exhibit lyotropic mesomorphism is of considerable interest for developing new target drug delivery systems.

Chitosan is a semi-synthetic biopolymer, which is widely used in a variety of biomedical applications due to its biocompatibility, biodegradability, and antibacterial activity [12]. Recent reviews [13,14] have summarized applications of chitosan as a component of drug delivery systems. Chitosan and chitosan-based materials demonstrate unique characteristics as drug delivery platforms because of their active primary amino groups that allow for their easy chemical modification. Simple and mild methods can be used for the encapsulation of biomolecules or drugs into chitosan. The mucoadhesion capabilities of chitosan facilitate its transport across mucosal barriers [15,16]. The surfaces of nanoparticles modified by chitosan can be easily transformed to provide required tumor targeting. Due to their tunable nanoscale and surface flexibility, such nanoparticles offer higher stability and are also compatible with a wide range of anti-cancer drugs [17]. Chitosan-based drug delivery hydrogels contain colloids and drugs embedded within their three-dimensional matrix. Such a matrix can play the role of a second barrier to the diffusion of drugs through the system, allowing for a better control of their release [18].

There are, however, few publications that describe the liquid crystalline behavior of chitosan in solvents. As shown by Prabaharan et al., solutions of chitosan in formic, acetic, and monochloroacetic acids may exhibit cholesteric mesophases at certain concentrations [19]. Cholesteric ordering in solutions of these acids is, however, a local effect that results in forming only domains of mesophases in isotropic liquids. Chang et al. describe the liquid crystalline behavior of chitosan in malic acid depending on its degree of deacetylation and molecular weight [20]. In our opinion, however, the data of polarized optical microscopy raise doubts on the effects of its birefringence. The experimental pictures demonstrate gels rather than lyomesophases. Kuse et al. studied the liquid crystal properties of highly substituted chitosan [21]. Its synthesis, however, is complicated and involves the use of a toxic precursor and toxic organic solvents. This compound, therefore, does not follow the requirements for drug delivery systems. A liquid crystal mucoadhesive precursor comprising 0.5% chitosan dispersion has been proposed for the disease treatment of vaginal candidiasis [22]. We have previously studied the lyotropic behavior of chitosan in solutions of acetic [23], succinic and, ascorbic acids [24]. The phase behavior of the systems was studied. The areas that formed gels and lyomesophases were found to depend on the concentration of acids and the content of chitosan.

In this work, we report on the synthesis of new biocompatible systems based on chitosan biopolymer and lactic acid (LA), the components involved in physiological processes, and evaluate the possibility of their application for the delivery of bioactive substances. A series of new Chit/LA LLCs was prepared. The correlations of liquid crystal properties, intermolecular interactions, and rheology of these systems were determined. To obtain stable LLC phases in broad concentration and temperature ranges, the following three-component system was prepared: chitosan–LA nonionic surfactant (C_12_EO_4_). Riboflavin is a hydrophilic compound that was used in this work to characterize the application potential of the studied lyotropic mesophases as drug delivery systems. Riboflavin participates in redox reactions and plays an important role in metabolic processes that involve carbohydrates, proteins, lipids, and the synthesis of hemoglobin [25,26]. In addition, riboflavin demonstrates a protective effect in various medical pathology conditions, such as sepsis, pellagra, acne vulgaris, and long-term non-healing ulcers. As shown by Darguzyre et al., riboflavin also reduces the risk of some forms of human cancer [27]. The skin permeability of riboflavin is predicted to be low, as its log*K*_o/w_ = 1.46 [28]. A lipophilic stratum corneum generally limits the skin’s permeability to hydrophilic drugs. Desai et al. [29] report that such drugs are incapable of a simultaneous sufficient permeation through skin to provide a pharmacological effect without skin irritation, even if chemical enhancers are added. In order to reveal the potential transdermal penetration capabilities of the studied delivery systems, we investigated biocompatible Chit/LA systems doped with riboflavin. Finally, the penetration of riboflavin through a hydrophobic membrane was simulated by the stratum corneum model. The viscosity data were correlated with the release kinetics. The proposed biocompatible systems are attractive as potential supramolecular platforms for transdermal delivery and controlled drug-release applications.

## 2. Results and Discussion

### 2.1. Liquid Crystal Properties of Chit/LA Binary Systems

LA is a suitable solvent for developing biocompatible transport systems because of its involvement in a variety of biochemical processes in the human body. To characterize the solubility of chitosan in LA and reveal the lyotropic mesomorphism conditions, we used polarized optical microscopy (POM). The concentrations of chitosan and LA were varied in the range of 5–16 wt. % and 2–15 wt. %, respectively. Figure 1 shows the images of the Chit/LA systems observed in polarized light.

Chitosan forms non-geometric textures in LA (Figure 1a). These textures are uncertain and it is difficult to determine the type of supramolecular organization in lyotropic mesophases by polarized optical microscopy. According to the literature data, it is challenging to obtain high-quality textures of polymer LLCs because of the intrinsically complex supramolecular organization of polymer systems [30]. Chitosan was found to form gels in a low concentration range of 2–13 wt. % (Figure 1b). These gels are represented by transparent phases, which take an intermediate position between an LLC phase and a homogeneous solution. A birefringence effect typical for LLC systems is not observed in the gels in polarized light. The concentration limits found for the studied gels and LLCs are shown in Figure 1c.

According to the POM data, the concentration limits of LA gelation are 2–7 wt. %. To obtain lyotropic mesophases, we need 10–15 wt. % of LA and 13–16 wt. % of chitosan.

### 2.2. Liquid Crystal Properties of Ternary C_12_EO_4_/(Chit:LA) Systems

To expand the concentration limits of lyotropic mesophases, the systems with added nonionic surfactant C_12_EO_4_ were developed. Previously, we studied the LLCs that consist of nonionic surfactants based on oligo(ethylene oxide) [31]. It was found that the systems that contain tetraethylene glycol monododecyl ether form a lamellar mesophase in water [32]. This phase can structurally mimic biological membranes. This surfactant was, therefore, used to obtain stable lyotropic biocompatible mesophases.

The POM images of C_12_EO_4_/(Chit:LA) samples demonstrate a typical texture with “oily streaks” that are characteristic features of a lamellar mesophase (Figure 2a,b).

Based on the POM data, the concentration and temperature ranges of the lyotropic mesophases were determined. The phase diagram of the C_12_EO_4/_(Chit:LA) system is provided in the Figure 2c.

Figure 2 shows that the lamellar mesophase forms in broad C_12_EO_4_ concentration limits of 20–80 wt. %. In this range, higher concentrations of the added surfactant reduce the temperature of the phase transition from the lamellar phase to the respective isotropic liquid. The maximum value T_ph.tr_ = 66.4 °C was observed for the C_12_EO_4_/(Chit:LA) system with 20/80 wt. % of chitosan and LA, respectively. In the ternary system, the two-phase region (a lamellar mesophase and an isotropic liquid) exists in the broad temperature range of 9–26 °C. Previously, we studied the liquid crystalline behavior of the C_12_EO_4_/Chit/acetic acid systems [23]. For these systems, the concentration range of the lamellar mesophase was found to be 55–80 wt. %. C_12_EO_4_. The two-phase state (coexistence of a lamellar mesophase and an isotropic liquid) emerged in the temperature range of 15–18 °C. The temperature of the phase transition to an isotropic liquid for C_12_EO_4_/Chit/acetic acid systems was found to be T_ph.tr_ = 62–64 °C, which is similar to the C_12_EO_4_/(Chit:LA) system.

Various organic acids exert a considerable influence on the concentration and temperature ranges of lyomesophases and their stability [24]. Among ascorbic, succinic, lactic, acetic, and citric acids, it is citric acid that provides the broadest concentration and temperature ranges of lyomesomorphism. In comparison with the previous results, the Chit/LA system is more stable, and its mesophase exists throughout a year.

### 2.3. FTIR Spectroscopy Studies

To characterize the molecular interactions between the LLC components, we recorded the FTIR spectra of the individual chitosan (Appendix A) and multicomponent systems.

The FTIR spectrum of the chitosan solution in LA is considerably different from those of individual substances. A broad band in the region of 2760–3712 cm^–1^ can be attributed to the OH-stretching mode of the LA solution (Figure 3a). In contrast to the spectrum of powdered chitosan (see Appendix A), an intense band (1645 cm^–1^) appears in the range of 1700–1500 cm^–1^. This band characterizes the antisymmetric bending of the ${\mathrm{NH}}_{3}^{+}$ group, which shifts by 10 cm^–1^ towards lower frequencies. In the spectrum of Chit/LA, there is no band at 1338 cm^–1^, which can be attributed to the ${\mathrm{NH}}_{3}^{+}$ symmetric bending [33,34]. Such changes indicate that electrostatic interactions occur between the ${\mathrm{NH}}_{3}^{+}$ groups and the lactate ions. The bands in the 1200–950 cm^−1^ range are attributed to the stretching modes of the C−C and C−O functional groups of LA [35]. The spectral changes at 1235 cm^–1^ and 1134 cm^–1^ are clear evidence of intermolecular interactions occurring in the Chit/LA system. Summarizing the results of the FTIR absorption experiments, we can conclude that the Chit/LA systems contain complexes in the form of the following ionic pairs: [ChitH^+^· CH_3_−CH(OH)−COO^−^] (see Figure 3a).

The FTIR spectrum of the LLC system in the presence of C_12_EO_4_ is also considerably different from those of the individual surfactants and solutions of chitosan in LA (Figure 3b). The bands of νOH (from 3473 to 3396 cm^−1^) and νCO (from 1121 to 1097 cm^−1^) shift to the lower frequency range of the C_12_EO_4_/(Chit:LA) spectrum compared with the spectrum of individual C_12_EO_4_ [23]. This effect indicates stronger hydrogen interactions occurring in the LLC system. We can assume that the formation of the C_12_EO_4_[ChitH^+^·CH_3_−CH(OH)−COO^−^] complex in the LLCs phase occurs through hydrogen bonding with the oxygen atoms of oxyethylene groups.

### 2.4. Rheological Properties

Viscosity is among the key properties of drug delivery systems. By varying the viscosities of such systems, we can control the release of bioactive substances. Higher viscosities can inhibit the release of water-soluble drugs. Lower viscosities are not favorable properties for transdermal applications of loaded drugs. On the other hand, they can intensify drug release. Figure 4 demonstrates the viscosity curves of the studied LLCs and gel systems.

The LLCs represented by Chit/LA and C_12_EO_4_/(Chit:LA) systems demonstrate high viscosities at low shear rates. This effect is associated with the presence of large structural units. At higher shear rates, molecularly organized LLC domains undergo compression and are easily oriented in the direction of the shear flow. Orientation causes a decrease in viscosity. The Chit/LA 5/95 wt. % gel has the lowest viscosity. Similar effects were revealed for gel systems of chitosan and collagen [36].

In general, LLCs demonstrate complex rheological responses under shear loads due to their microstructural reorganization. We have previously shown that shear deformations affect the structure and orientation of mesophases, especially for lamellar phases [24].

In order to determine the flow models of the studied gels and LLCs, the plots of shear rates versus shear stresses were analyzed (Figure 4). The non-Newtonian pseudoplastic flow mode was observed in all the systems. Previously, a similar behavior was reported for the LLCs based on C_12_EO_4_/H_2_O and C_12_EO_10_/H_2_O [31], for the Chit/acetic acid/C_12_EO_4_ system [23] and the κ-carrageenan/H_2_O/C_12_EO_n_ system [37].

For a more accurate characterization of the rheological behavior, the flow curves (Figure 5) were subdivided into two sections corresponding to smaller $\dot{\gamma }$ = 0.07–7.5 s^−1^ and higher $\dot{\gamma }$ = 7.5–90 s^−1^ shear rates.

Rheological curves were correlated using the Bingham, Power Law, Herschel–Bulkley, and Casson models (Appendix A).

In general, the rheological behavior of pseudoplastic and viscoplastic systems can be described by various microrheological models [38]. A Bingham model is applicable to surfactant–water mixtures with a yield stress [39]. However, this model approximates only a small part of the flow curve at high shear rates [40]. In Ref. [41], a non-Newtonian flow of lyomesophases was described by the Power Law model. This model is suitable for describing the rheological behavior of gels [42] and chitosan-based gels with carbon nanotubes [43]. Since Chit/LA LLCs are characterized by a yield stress τ_0_, this model was not considered in our study (Appendix A). The Herschel–Bulkley model provides an adequate simulation of flow curves but offers no respective microrheological justification [38]. Using this model, we failed to calculate a positive yield strength for the Chit/LA gel, which was the reason for excluding it from the list of applicable models (Appendix A).

According to the equations (Equation (S1)–(S5)), the Casson model provides the best fitting degree for binary and ternary LLCs (Table 1), as we can see from their respective R^2^ values of 0.970 and 0.966. The microrheological Casson model is based on the flow of chain-like and rod-like aggregates capable of orienting in the shear field [44]. The Casson model is suitable for simulating the rheological behavior of LLC mesophases. The coefficients of its equations are found to be different for LLC phases at low and high shear rates. These systems, therefore, demonstrate a complex rheological behavior.

The Casson model provides a better degree of fitting at lower shear rates for the Chit/LA gel. At higher shear rates, the best degree of fitting is provided by the Power Law model (Appendix A). This result may be associated with unfolding and ordering of polysaccharide chains along the shear field.

### 2.5. The Study of Riboflavin Release from Gels and LLC Systems

Controlled drug delivery systems are advanced carriers for transporting pharmaceutical compounds. LLCs have great potential as drug delivery systems due to their set of relevant mesophase geometries and physicochemical properties, such as bioavailability, and their ability to carry a range of water-insoluble drugs to perform their sustainable release.

The kinetic curves of riboflavin released from the LLCs and the gel diffused through the membrane are shown in Figure 6. The values of the release time and the rate of the process are summarized in Table 2.

According to Table 2, riboflavin demonstrates a complex release behavior. For the ternary LLCs C_12_EO_4_/(Chit:LA) system, a rapid release is observed within 45 min. This system does not have an induction period. The diffusion release starts immediately after the system makes contact with a liquid medium. The kinetic curves show two distinct sections with different rates (dQ/dt). The first section is the interval of 0–23 min with a higher release rate of 3.4 mmol·cm^−2^min^−1^. In the second time interval, the release rate slows down to 1.0 mmol·cm^−2^min^−1^.

As compared with the previously studied C_12_EO_4_/Chit/acetic acid system [24] as a carrier for rhodamine (t_release_ = 180 min), the C_12_EO_4_/(Chit:LA) system is characterized by a zero-induction period and faster release dynamics.

The studied Chit/LA gel offers a longer release time as compared with the respective ternary system (Figure 6). This system is also characterized by two periods with different release rates. In the time interval of 51 ≤ t ≤ 80, a higher rate of 1.7 mmol·cm^−2^·min^−1^ is observed. In the time interval of 167 ≤ t ≤ 206, a slower rate of 0.5 mmol·cm^−2^·min^−1^ is observed. The system gradually reaches its kinetic plateau in 210 min.

The maximum release time was observed for the binary LLCs Chit/LA system. This system is characterized by a small induction period in the initial region: only 7 mmol·cm^−2^ of riboflavin releases in 58 min. The release of the bioactive substance completes in 540 min. The resulting release rate is 0.8 mmol·cm^−2^·min^−1^ in the interval of 7 ≤ t ≤ 543 min.

Therapeutic applications require delivery systems that provide both prolonged and fast-release capabilities. Drug diffusion kinetics depend on a variety of factors, such as the geometry, polarities, and molecular weights of drugs, as well as the internal structure of a drug-loading matrix. As shown in Ref. [45], the release rates of loaded drugs are higher in lamellar mesophases as compared with inverse hexagonal and cubic phases. These results agree with our experiments, showing that the lamellar LLCs C_12_EO_4_/(Chit:LA) system demonstrates the fastest release dynamics. According to Ref. [46], the release of drugs from LLCs can be adequately described by the Higuchi diffusion kinetics model. The riboflavin release profiles from LLCs and the gels were plotted against the square root of time (Appendix A). These results prove that the release dynamics of liquid crystalline systems can be described in terms of Fickian diffusion.

### 2.6. Model of Riboflavin Penetration through a Hydrophobic Membrane

According to Ref. [47], the surface morphology and internal structure of a polytetrafluoroethylene (PTFE) membrane can be represented by oriented micropores separated by fibrils. A characteristic feature of micropores is their spatial periodicity. The kinetics of riboflavin release is found to depend on several factors, such as the structure of the system, the rheological characteristics of the LLCs, and the rate of riboflavin penetration through the membrane pores (Figure 7).

For the LLCs represented by the Chit/LA system, higher viscosities were observed at lower shear rates. This effect is associated with the presence of large molecularly organized aggregates. This explains the slow riboflavin releases from binary LLCs systems that reach 540 min. The maximum release amount of riboflavin is about 98%. Due to the molecularly organized viscous structure of the LLC phase, polymer macromolecules do not clog the pores of the membrane and, therefore, provide a nearly complete release of riboflavin. The Chit/LA gel releases only 42% of riboflavin in 210 min. A possible reason is the effect of polymer macromolecules that are not bound by intermolecular bonds. Such macromolecules partially clog the pores over time and prevent riboflavin from completing release. The ternary C_12_EO_4_/(Chit:LA) LLCs system has the lowest viscosity due to its lamellar structure. This structure favors the penetration of surfactants and polymer macromolecules into the pores of the membrane and reduces the amount of the released riboflavin to 17% in 47 min.

## 3. Materials and Methods

### 3.1. Materials

Chitosan (product of Iceland from shrimp shells) was purchased from Sigma-Aldrich (St. Louis, MO, USA) and additionally characterized. The degree of deacetylation of chitosan was found to be 95 ± 0.8% according to FTIR. The average molecular weight was found to be 108 kDa according to dynamic light scattering. Tetraethylene glycol monododecyl ether was purchased from Sigma-Aldrich (USA) and used as received. Lactic acid (80% aqueous solution) was purchased from Merck (Darmstadt, Germany) and used as received. Riboflavin was purchased from BASF (Ludwigshafen, Germany) and used as received. Double-distilled water was used in all experiments.

### 3.2. Preparation of Gels and Llcs

Chitosan solutions were prepared by complete mixing of polysaccharide samples in the 2–15 wt. % range of LA concentrations. To prepare LA solutions, we used double-distilled water.

To prepare ternary systems, a pre-calculated amount of C_12_EO_4_ was added to the chitosan solution in lactic acid. For example, to synthesize the LLC system C_12_EO_4_/(Chit:LA), we mixed 50/50 wt. %, 0.2 g of chitosan with 3.8 g LA (the concentration of LA aqueous solution C_LA_ = 2%) and then added 4 g of C_12_EO_4_. To plot the phase diagram, we used the constant weight ratio of Chit:LA = 5:95 wt. %. The LLC systems were aged for 14 days at 25 °C before use.

To homogenize all the systems, we centrifuged them (8000 rpm at 25–30 °C) for 30 min. The systems were then aged for 14 days at 25 °C before use.

To prepare systems with riboflavin, a pre-calculated volume of riboflavin aqueous–alcohol solution (50 vol.% alcohol, C_riboflavin_ = 1×10^−3^ M) was added to the LLC phase. The weight ratio of this solution and the LLC phase was set to 1:3 in all the experiments. The samples were centrifuged (8000 rpm at 25–30 °C) for 30 min to obtain a completely homogenous system.

### 3.3. Polarized Optical Microscopy (POM)

The type of a liquid crystal phase and the interval of mesophase existence were determined by Olympus BX51 polarized light microscope equipped with Linkam precision heating system (Olympus Life Science Europa GmbH, Hamburg, Germany). The samples were heated from 20 to 70 °C at a rate below 5 °C/min. Phase transition temperatures were measured with an accuracy of ±0.1%.

### 3.4. Determination of Viscosity

The measurements were carried out by a Brookfield DV-II + Pro rotational viscometer (Brookfield Engineering Lab. Inc., Middleborough, MA, USA) (equipped with the Rheocal 32 software) with a cone/plate measuring system, cone angle 0.8 deg, and cone radius 2.4 cm. The measurement accuracy was within ±1.0% of its full range and a spring torque of 673.7 dyne-cm. The stationary plate forms the bottom of a sample and can be removed to fill 0.5 mL of a studied sample. The C-40 spindle was used in this viscometer. It is equipped with an electronic gap setting for a faster set-up, minimizing the possibility of gap adjustment errors. The experiments were carried out at 25 °C. The bath with the computer temperature control system was used in all the viscosity experiments. Each viscosity measurement was performed at least three times. The standard deviation was 2%.

### 3.5. FTIR Spectroscopy

FTIR spectra were recorded by a Bruker Alfa spectrometer (Bruker Optik GmbH, Ettlingen, Germany) using the ATR module with a germanium window. The experiments were performed in the 4000–400 cm^−1^ frequency range at atmospheric pressure and 25 °C temperature. 

### 3.6. In Vitro Riboflavin Release

The rate of riboflavin mass transfer from the gels and LLCs was measured in Franz glass diffusion cells consisting of donor and acceptor compartments separated by a membrane. MF-Millipore membranes (pore size 0.45 μm, membrane diameter 13 mm) made of biologically inert polytetrafluoroethylene were used. Membranes of this type are suitable for determining the rate of drug release [48,49]. The receiving medium was phosphate buffer (pH = 7.4). 

The concentration of riboflavin in the receiving medium was measured by Lambda 35 UV/VIS Spectrometer (PerkinElmer Ltd., Buckinghamshire, UK). To perform a measurement, a sample of solution (3 cm^3^) was taken and transferred to a spectrophotometric cuvette and returned to the acceptor part of the Franz cell immediately after the measurement was complete [50].

The concentration values were calculated using Equation (1):C = A_max_/ε_max_·l,(1)
where A is the optical density at λ_max_, l is the width of the cell, and ε_max_ is the molar extinction coefficient that was determined from the linear calibration curve of riboflavin (ε = 6743.7 M^−1^cm^−1^ at λ = 264 nm).

The amount of substance Q, which diffuses through the unit area of the membrane surface over the time t, is calculated by Equation (2):Q = C·V/S_0_,(2)
where C is the experimental concentration of the target component in the receiving medium at the time t, V is the volume of the receiving medium (6 cm^3^), and S_0_ is the diffusion area that is equal to the area of the Franz cell opening (0.64 cm^2^).

The experiments were repeated three times; then, the average Q values were calculated. From the dependencies Q(t), the diffusion mass transfer rates of riboflavin were determined. All the experimental data were obtained at 25 ± 1 °C. The drug-release test was conducted at least three times. The standard deviation was 3%.

### 3.7. Statistical Analysis

The statistical analysis of the rheological parameters and release dynamics was performed using standard methods. The ANOVA method was used to calculate the significance (*p* < 0.05).

## 4. Conclusions

This work describes a simple method for preparing biocompatible LLCs. The solubility and liquid crystalline properties of systems based on chitosan and LA were studied. The concentration ranges of gelation and the existence of the LLC phases were found. Chitosan forms gels in its low concentration range of 2–13 wt. %. The concentration limits of LA gelation are 2–7 wt. %. To obtain lyotropic mesophases, 10–15 wt. % of LA and 13–16 wt. % of chitosan were needed. The LLC phase emerged through the hydrogen bonding occurring in the ionic complex [ChitH^+^·CH_3_-CH(OH)-COO^−^]. An in vitro release study revealed that binary Chit/LA LLCs systems are capable of providing a sustained release of riboflavin in comparison with gels or ternary delivery systems. A correlation between the plastic viscosity of chitosan-based systems, the release dynamics of riboflavin, and the type of penetration through a hydrophobic PTFE membrane was established. Assuming that the behavior of a PTFE membrane can be described by the model of the stratum corneum, we have shown that a high viscosity of the binary Chit/LA LLCs prevents the clogging of pores that results in a prolonged and complete release of riboflavin. Distinct composition–property relationships were found for the studied LLCs, which can be applied to developing transdermal drug delivery systems for pharmacological applications. We assume that Chit/LA LLCs are able to encapsulate a wide range of bioactive compounds and, therefore, offer various possible routes of administration, such as transdermal, oral, topical, or vaginal. These studies will be a matter of further research. Thus, the results of this work offer a simple method for the synthesis of stable and biocompatible LLC-based drug delivery systems with controlled-release capabilities. Such systems represent a promising and original approach to developing nanoscale controlled drug delivery vehicles.

## Figures and Tables

**Figure 1 ijms-23-13207-f001:**
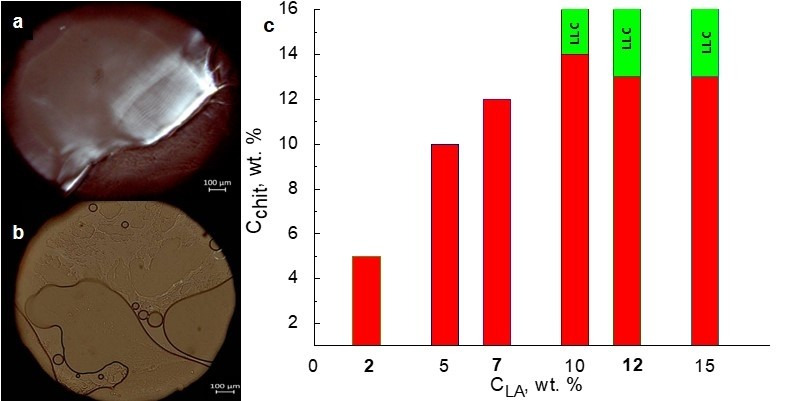
POM images: (**a**) Chit/LA 15/85 wt. % (C_LA_ = 10 wt. %); (**b**) Chit/LA 8/92 wt. % (C_LA_ = 2 wt. %); (**c**) Diagram of chitosan solubility in LA.

**Figure 2 ijms-23-13207-f002:**
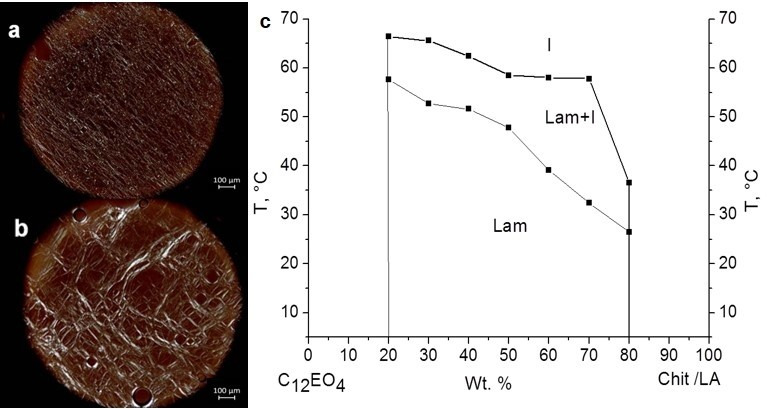
POM images of C_12_EO_4_/(Chit:LA) at C_12_EO_4_: (**a**) 70 wt. %; (**b**) 30 wt. %; (**c**) phase diagram of the C_12_EO_4/_(Chit:LA).

**Figure 3 ijms-23-13207-f003:**
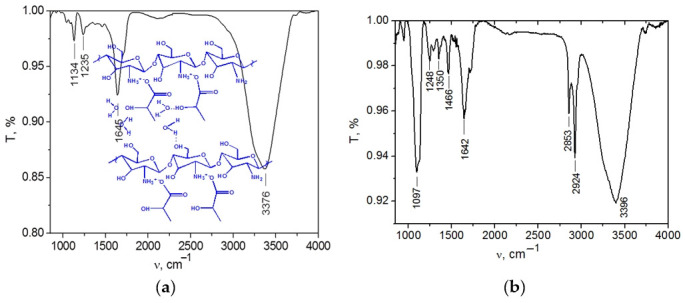
FTIR spectra of Chit/LA 5/95 wt. % (**a**); C_12_EO_4_/(Chit:LA) 50/50 wt. % (**b**).

**Figure 4 ijms-23-13207-f004:**
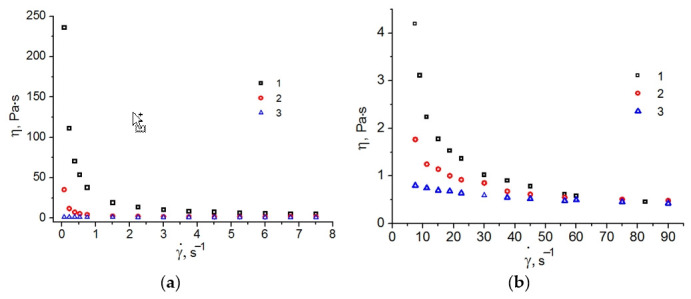
Viscosity curves at low (**a**) and high (**b**) shear rates of the following systems: 1. Chit/LA 15/85 wt. %, 2. C_12_EO_4/_(Chit:LA) 50/50 wt. %, 3. Chit/LA 5/95 wt. %.

**Figure 5 ijms-23-13207-f005:**
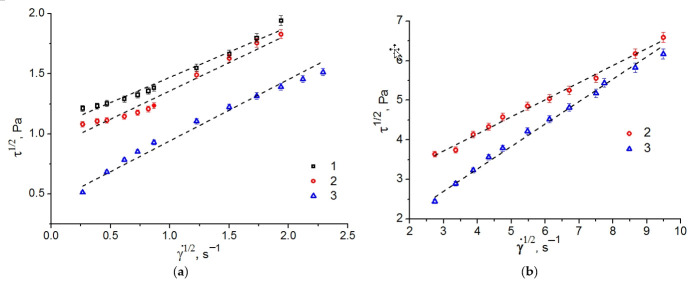
Flow curves in square root coordinates at low (**a**) and high (**b**) shear rates of the systems: 1. Chit/LA 15/85 wt. %, 2. C_12_EO_4_/(Chit:LA) 50/50 wt. %, 3. Chit/LA 5/95 wt. %. Dashed lines (--) are plotted according to the Casson model.

**Figure 6 ijms-23-13207-f006:**
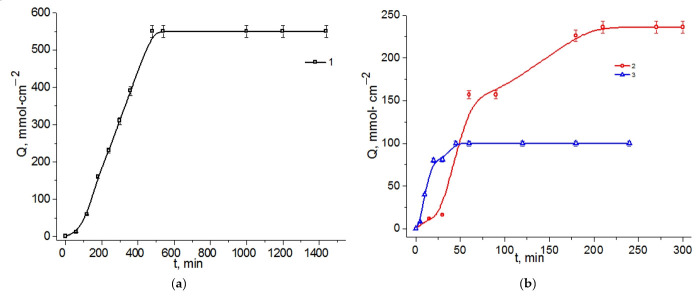
Release profiles of riboflavin from (**a**) Chit/LA 15/85 wt. %—1; (**b**) Chit/LA 5/95 wt. % (C_LA_ = 2 wt. %)—2, C_12_EO_4_/(Chit:LA) 50/50 wt. %—3.

**Figure 7 ijms-23-13207-f007:**
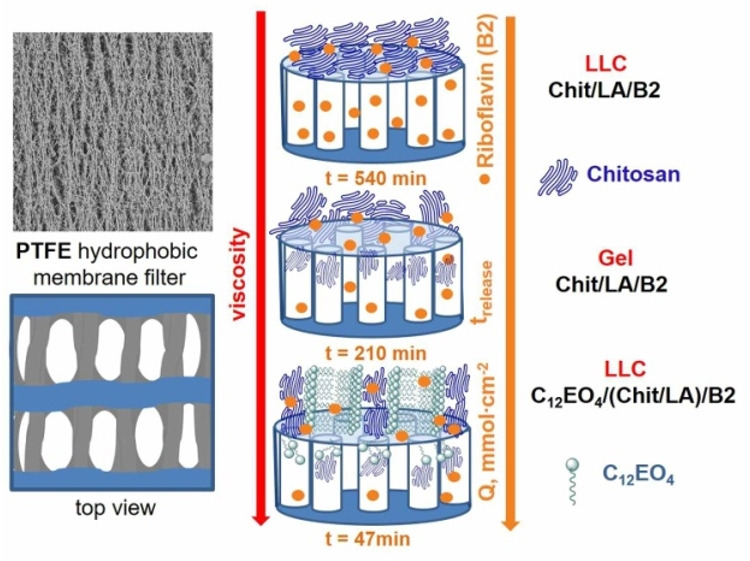
Scheme of riboflavin penetration through a PTFE membrane from gel and LLC systems.

**Table 1 ijms-23-13207-t001:** Rheological characteristics of the systems at smaller $\dot{\gamma }$ = 0.07–7.5 s^–1^ and higher $\dot{\gamma }$ = 7.5–90 s^–1^ shear rates (Casson model).

System	$\dot{\gamma }$ = 0.07–7.5 s^−1^			$\dot{\gamma }$ = 7.5–90 s^−1^		
	μ, mPa·s	τ_0_, Pa	R, %	μ, mPa·s	τ_0_, Pa	R, %
C_12_EO_4_/(Chit:LA) LCC	556.0	1.38	97.0	181.6	6.24	97.9
Chit/LA Gel	754.7	0.02	96.6	252.4	1.80	96.9
Chit/LA LCC	5190.0	18.17	98.1	298.7	1.25	95.9

Where R is the correlation coefficient of experimental and calculated data, *p* < 0.05 vs. C_12_EO_4._

**Table 2 ijms-23-13207-t002:** Parameters of riboflavin mass transfer.

System	Induction Period	t, min	dQ/dt
C_12_EO_4/_(Chit: LA) LLC	-	47	3.4 (0 ≤ t ≤ 23) 1 (25 ≤ t ≤ 45)
Chit/LA gel	+	210	1.7 (51 ≤ t ≤ 80) 0.5 (167 ≤ t ≤ 206)
Chit/LA LLC	+	540	1 (7 ≤ t ≤ 543)

*p* < 0.05 vs. C_12_EO_4._

## Data Availability

The data presented in this study are available on request from the corresponding author: Yuriy G. Galyametdinov.

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
