# Peer review of "Chitosan/Lactic Acid Systems: Liquid Crystalline Behavior, Rheological Properties, and Riboflavin Release In Vitro"

_ijms, 2022, doi:10.3390/ijms232113207_

Round 1

Reviewer 1 Report

After reading the manuscript, I consider that can be published as it is. The data are well presented and analysed.

I have only one question regarding the introduction of riboflavin in system. The authors mentioned in the experimental part that they used ethanol/water mixture to prepare riboflavin solution. What is the influence of the ethanol on the properties of the chitosan system?

Author Response

Response to Reviewer 1:

Thank you for review of our manuscript.

Point 1. I have only one question regarding the introduction of riboflavin in system. The authors mentioned in the experimental part that they used ethanol/water mixture to prepare riboflavin solution. What is the influence of the ethanol on the properties of the chitosan system?

Response 1:

According to the literature data, the addition of ethanol affects the polyelectrolyte swelling of chitosan in aqueous solutions, especially if chitosan is in the salt form. In this case, the conformational state of chitosan in solution, as well as its solubility and viscosity will change [1,2].

In our case, chitosan is in the protonated form, additionally associated with lactic acid molecules. In LLC systems the concentration of chitosan is high and the total content of ethanol in the system is low. So, we think that the effect of ethanol will be minimal.

  1. Shilova, S.V.; Mirgaleev, G.M.; Tretyakova, A.Y.; Barabanov V.P. Polyelectrolyte complexes of chitosan with sodium carboxymethyl cellulose in water–alcohol media and microcapsules based on them. Polym. Sci. – A, 2020, 62, 630-635. http://dx.doi.org/10.1134/S0965545X20050156
  2. Uspenskii, S.A., Vikhoreva, G.A., Sonina, A.N., Galbraikh, L.S. Properties of acetic-acid alcohol-containing solutions of chitosan. Fibre Chemistry, 2010, 42, 88-91. http://dx.doi.org/10.1134/S0965545X20050156

Reviewer 2 Report

Specific comments:

1.       Throughout the text: I don't really understand what the authors mean by the 5/95 wt.% ratio (CLA = 2 wt.%), much less the C12EO4/(Chit/LA) 50/50 wt.% ratio etc. Please clarify the concentrations of each component in all solutions used.

2.       Throughout the text: The name Chit+ is not quite accurate, it lacks a proton - it should be, for example, ChitH+.

  1. Check once again the appropriateness of using acronyms. Overuse or inappropriate use of acronyms makes reading paper difficult. Please note the following regarding acronyms:

Acronyms should be spelled out upon first use, followed by the acronym itself in parentheses.

Subsequently, only the acronym should be used in the text.

To keep the use of acronyms to a minimum, only insert an acronym if the term is used at least three times.

4.       There is a significant number of unreasonable self-citations in the paper. I understand the desire of the authors to promote their own work; however, the scientific community is opposing the massive self-citations. I would recommend the authors to revise the references and leave only those needed for discussion. Nature published an article discussing this issue https://www.nature.com/articles/d41586-019-02479-7.

5.       In order to give readers, the maximum appreciation of how your work builds on previous results, EACH ONE of the cited sources should be discussed individually and explicitly to demonstrate their significance to your study. I ask that you use the authors' surnames as the SUBJECT of a VERB, and then state in one or two sentences what they claim, what evidence they provide to support their claim, and how you evaluate their work. I also, therefore, ask that you avoid citing more than one reference in one sentence. This will give you a chance to discuss each reference separately.

What I am asking for is something like: "Selivanova et al. describes the development…" For example, would not accept massive citations like [1-3], [4-7], [8-12], [13-16], [17-23] etc.

6.       Lines 28-29: Remove “In the”.

7.       Line 38: Strictly speaking, chitosan is not natural, but semi-synthetic polymer. Please correct.

8.       Lines 157-159: In “NH3+” 3 must be subscript, and + superscript (three times).

9.       Line 315: The chitosan sample must be thoroughly characterized regarding its molecular weight (by viscometry, light scattering, or size exclusion chromatography) and the degree of deacetylation (by NMR, IR, elemental analysis, or titration). Also, indicate the source of chitosan (crab, shrimp, fungi, etc.). The properties of chitosan are very dependent on these parameters; therefore, the wide ranges of values provided by the manufacturer are clearly insufficient. When working with natural polymers, be prepared to characterize in detail every and each sample.

10.   Line 383-384: Reproducibility is usually characterized by the standard deviation. What does 97% reproducibility mean?

Author Response

Response to Reviewer 2:

Thank you for review of our manuscript.

According to reviewer comments the references were changed, self-citation percentage was reduced. Introduction was rewrite. The relevant references were included into introduction and format reference citation has been changed. The section Materials and Methods was improved. English language was checked and polished.

Point 1.       Throughout the text: I don't really understand what the authors mean by the 5/95 wt.% ratio (CLA = 2 wt.%), much less the C12EO4/(Chit/LA) 50/50 wt.% ratio etc. Please clarify the concentrations of each component in all solutions used.

Response 1:  To prepare the ternary systems, a pre-calculated amount of C12EO4 was added to the chitosan solution in lactic acid. For example, to synthesize the LLC system С12EO4/(Chit:LA) 50/50 wt.%, 0.2 g of Chit was initially mixed with 3.8 g LA (the concentration of LA aqueous solution СLA = 2 %), then 4 g of С12EO4 was added. To plot the phase diagram, we kept constant the mass ratio Chit:LA of 5:95 wt.%.

Point 2.       Throughout the text: The name Chit+ is not quite accurate, it lacks a proton - it should be, for example, ChitH+.

Response 2:

Corrected.

Point 3. Check once again the appropriateness of using acronyms. Overuse or inappropriate use of acronyms makes reading paper difficult. Please note the following regarding acronyms:

Acronyms should be spelled out upon first use, followed by the acronym itself in parentheses.

Subsequently, only the acronym should be used in the text.

To keep the use of acronyms to a minimum, only insert an acronym if the term is used at least three times.

Response 3:

Using of acronyms were checked and corrected throughout the text.

In the main text of the manuscript, we addressed the comment of the Reviewer and used only the original term “chitosan” without introducing its acronym to improve the readability of the revised manuscript. We used the acronym “Chit” to identify only the LLC complexes in the respective chemical abbreviations, for example:

3.2. Liquid crystal properties of ternary С12EO4/(Chit:LA) systems

Point 4.       There is a significant number of unreasonable self-citations in the paper. I understand the desire of the authors to promote their own work; however, the scientific community is opposing the massive self-citations. I would recommend the authors to revise the references and leave only those needed for discussion. Nature published an article discussing this issue https://www.nature.com/articles/d41586-019-02479-7.

Response 4:

The self-citation percentage in the paper is 16%. According to the mentioned publication https://www.nature.com/articles/d41586-019-02479-7, “Those with greater than 25% self-citation are not necessarily engaging in unethical behaviour, but closer scrutiny may be needed” this is the allowable percentage.

We have added relevant comments to the text. Some references have been removed.

Point 5.       In order to give readers, the maximum appreciation of how your work builds on previous results, EACH ONE of the cited sources should be discussed individually and explicitly to demonstrate their significance to your study. I ask that you use the authors' surnames as the SUBJECT of a VERB, and then state in one or two sentences what they claim, what evidence they provide to support their claim, and how you evaluate their work. I also, therefore, ask that you avoid citing more than one reference in one sentence. This will give you a chance to discuss each reference separately.

What I am asking for is something like: "Selivanova et al. describes the development…" For example, would not accept massive citations like [1-3], [4-7], [8-12], [13-16], [17-23] etc.

Response 5:

The way of citations like [1-3], [4-7] are normal for Chemical journals. Nevertheless, we include key words for each references and corrected the format of the references. The Introduction was corrected.

Point 6.       Lines 28-29: Remove “In the”.

Response 1:

Removed

Point 7.       Line 38: Strictly speaking, chitosan is not natural, but semi-synthetic polymer. Please correct.

Response 7:

Corrected

Point 8.       Lines 157-159: In “NH3+” 3 must be subscript, and + superscript (three times).

Response 8:

Corrected in the text

Point 9.       Line 315: The chitosan sample must be thoroughly characterized regarding its molecular weight (by viscometry, light scattering, or size exclusion chromatography) and the degree of deacetylation (by NMR, IR, elemental analysis, or titration). Also, indicate the source of chitosan (crab, shrimp, fungi, etc.). The properties of chitosan are very dependent on these parameters; therefore, the wide ranges of values provided by the manufacturer are clearly insufficient. When working with natural polymers, be prepared to characterize in detail every and each sample.

Response 9:

Chit which we used in our experiments is a commercial product of Sigma-Aldrich (USA). The site of this company contains the NMR, IR and some other characteristics of the Chit. (CAS Number: 9012-76-4, MDL number: MFCD00161512, 448869 Sigma-Aldrich, https://www.sigmaaldrich.com/RU/en/product/aldrich/448869).

Point 10.   Line 383-384: Reproducibility is usually characterized by the standard deviation. What does 97% reproducibility mean?

Response 10:

Error has been fixed.

The standard deviation was 2% for the viscosity measurements.

The standard deviation was 3% for the riboflavin release measurement results

Round 2

Reviewer 2 Report

I strongly disagree with the authors on the need to characterize the chitosan sample. Most of us involved in "chitin science" use commercial samples of chitosan, but we know that the characteristics (molecular weight, dispersity, degree of deacetylation) of these commercial samples vary greatly from batch to batch. Numerous papers on chitosan show that the properties of this polysaccharide are very dependent on its characteristics, so the European Chitin Society, which is affiliated with IJMS, does not recommend the publication of results on chitosan without explicitely indicating its characteristics. Otherwise, the evaluation of any properties of chitosan becomes meaningless.

At least three characteristics for chitosan must be specified or determined with sufficient accuracy:

1) the source of chitosan (crab, shrimp, fungi, etc.). 

2) molecular weight (by viscometry, light scattering, or size exclusion chromatography) 

3) degree of deacetylation (by NMR, IR, elemental analysis, or titration). 

Author Response

Response to Reviewer 2:

Thank you for review of our manuscript.

Point 1. I strongly disagree with the authors on the need to characterize the chitosan sample. Most of us involved in "chitin science" use commercial samples of chitosan, but we know that the characteristics (molecular weight, dispersity, degree of deacetylation) of these commercial samples vary greatly from batch to batch. Numerous papers on chitosan show that the properties of this polysaccharide are very dependent on its characteristics, so the European Chitin Society, which is affiliated with IJMS, does not recommend the publication of results on chitosan without explicitely indicating its characteristics. Otherwise, the evaluation of any properties of chitosan becomes meaningless.

At least three characteristics for chitosan must be specified or determined with sufficient accuracy:

1) the source of chitosan (crab, shrimp, fungi, etc.). 

2) molecular weight (by viscometry, light scattering, or size exclusion chromatography) 

3) degree of deacetylation (by NMR, IR, elemental analysis, or titration). 

Response 1: 

Considering your comments, we have supplemented the following characteristics of chitosan:

1) Chitosan was product of Iceland from shrimp shells.

2) The average molecular weight of chitosan was 108 kDa determined by using dynamic light scattering (Malvern Instruments Ltd., Worcestershire, UK)

The studies were carried out on a Zetasizer Nano ZS apparatus with a helium– neon laser (633 nm, 4 mW) from (Malvern Instruments Ltd., Worcestershire, UK).

0.1 M HCl with 0.05M KCl was used as a solvent. Before measurement, the solvent was filtered through a Millipore hydrophilic filter in the Millex HV Filter Unit with 0.45 µm pores. The following concentrations of chitosan solutions were prepared: 11.25, 5.625, 2.813, 1.469, 0.703 g/L. Light scattering intensity of the solvent was measured initially. Rayleigh scattering was assumed for molecular weight measurements. The software reduced the Rayleigh equation to a linear form from which a Debye plot was generated to show the variation in the average intensity versus concentration. The intercept corresponding to zero concentration was determined and then used to calculate the average molecular weight of polymers (Fig. 1). Instrument settings were 25°C at a scattering angle of 173°. Additionally, the refractive index increment was measured Δn/Δc  = 0.147 mg/L.

Figure 1. Debye plot for chitosan solutions

Molecular weight of chitosan 108 ±11,5 (kDa), correlation coefficient R2 = 0,997

Second virial coefficients A2  = 0.0295 ± 9.56e-05

3) The degree of deacetylation of chitosan was determination by IR spectroscopy.

The studies were carried out on a IR Affinity spectrophotometer (Shimadzu, Japan) with an ATR) attachment (Pike Technologies, USA). Spectra measurement range 700 – 4000 cm-1, spectral resolution 4.0 cm-1, number of accumulations 128. The device was purged with dry nitrogen to remove interfering water vapors.

For sample preparation dry powder of chitosan was ground in a ball mill to a finely dispersed state. The resulting powder was washed with deionized water until the absence of absorption bands of foreign substances in the washout spectrum. To 10 mg of the powder, 200 μL of deionized water was added and titrated with 0.1 M HCl to pH 4.5, which led to the dissolution of the sample with the formation of a viscous solution. 20 μl of the solution was placed on the surface of a ZnSe ATR sensitive element and the spectrum was recorded. The spectrum of water was recorded separately, which was then subtracted from the spectrum of the sample. The degree of deacetylation was determined according to [1] based on the spectra of calibration mixtures of N-acetylglucosamine and D-glucosamine.

The calculated degree of deacetylation DD = 95±0.8%.

  1. Jan Ken D Dimzon, Thomas P Knepper. Degree of deacetylation of chitosan by infrared spectroscopy and partial least squares // Int J Biol Macromol 2015, 72:939-45. doi: 10.1016/j.ijbiomac.2014.09.050.

Round 3

Reviewer 2 Report

I appreciate the work done by the authors to improve the manuscript. I suppose it can be published in its present form.

Author Response

Thank you for review of our manuscript. According your comment “Moderate English changes required” the manuscript was revised by a fluent English speaker. He published several papers in MDPI journals and was invited as a Guest Editor to a Q1 journal of MDPI. He performed a thorough proofreading of the final manuscript text and marked all the changes in red.